

# Development of longitudinal dunes under Pangaean atmospheric circulation

Hiroki Shozaki[1,2,*], Hitoshi Hasegawa[3,*]

[1] Deptartment of Earth and Planetary Science, Tokyo Institute of Technology, Tokyo, 152-8550, Japan
[2] Earth-Life Science Institute, Tokyo Institute of Technology, Tokyo, 152-8550, Japan
[3] Faculty of Science and Technology, Kochi University, Kochi 780-8520, Japan

[*]These authors contributed equally to this work.
*Correspondence to*: Hiroki Shozaki (hi.shozaki@elsi.jp) and Hitoshi Hasegawa (hito_hase@kochi-u.ac.jp)

**Abstract.** As a result of the large difference in heat capacity between land and ocean, global climate and
atmospheric circulation patterns in the supercontinent Pangaea were significantly different from today.
Modelling experiments have suggested the seasonal overturning of cross-equatorial Hadley circulation; however,
there are large discrepancies between model-generated surface wind patterns and the reported palaeo-wind
directions from aeolian dune records. Here, we present the  results of measurements of spatial distribution of
dune slip-face azimuths recorded in Lower Jurassic aeolian sandstones over a wide area of the western United
States (palaeolatitude: ~19°–27°N). The azimuth data of dune slip-faces reveal a bi-directional and oblique
angular pattern that resembles the internal structures of modern longitudinal dunes. Based on the spatial pattern
of slip-face directions and outcrop evidences, we suggest most of Lower Jurassic aeolian sandstones to be NNE–
SSW- to NNW–SSE- oriented longitudinal dunes, which likely formed as the result of a combination of westerly,
northwesterly, and northeasterly palaeo-winds. The reconstructed palaeo-wind pattern at ~19°–27°N appears to
be consistent with the model-generated surface wind pattern and its seasonal turnover.  The reconstructed palaeo-
wind patterns also suggest an influence of orbitally induced changes in atmospheric pressure configuration in
Pangaea.

## 1 Introduction

From the Carboniferous to the Jurassic, the supercontinent Pangaea dominated Earth. Because Pangaea was the
largest pole-to-pole landmass in Earth's history, atmospheric circulation during this period is thought to have
differed substantially from that of today (Kutzbach and Gallimore, 1989; Parrish, 1993). Modelling experiments
have indicated that Pangaean atmospheric circulation was characterised by cross-equatorial Hadley circulation
and large seasonal movement of the intertropical convergence zone (ITCZ) to near 30° on land in both
hemispheres (Kutzbach and Gallimore, 1989; Parrish, 1993; Rowe et al., 2007). Some studies have also
suggested significant changes in atmospheric pressure patterns in Pangaea caused by orbital-scale changes in the
seasonal and latitudinal distribution of solar radiation (Kutzbach, 1994; Winguth and Winguth, 2013). However,



the terrestrial environmental response to such seasonal- and orbital-scale changes in atmospheric pressure patterns during this period remains largely uncertain.

The aeolian dune record provides significant information of surface prevailing wind regime and atmospheric
circulation patterns in the past ( Lancaster, 1981; Parrish and Peterson, 1988; Peterson, 1988; Livingstone, 1989; Lancaster, 1990; Kocurek, 1991; Scherer, 2000; Lancaster et al., 2002; Loope et al., 2004; Beveridge et al., 2006; Sridhar et al., 2006; Rodríguez-López et al., 2008; Hasegawa et al., 2012). Modern deserts are generally developed in the subtropical high-pressure belt as a result of downwelling of the Hadley circulation, and aeolian dunes in desert areas record the prevailing surface wind pattern (e.g., trade winds and westerlies) in the form of
large-scale cross-beds (Breed et al., 1979; Lancaster, 1981; Wasson et al., 1988; Hesse, 2010; Hasegawa et al., 2012). In addition, the patterns of dune alignment and morphology have been considered to reflect the prevailing wind regime and mesoscale circulation pattern with its seasonal and long-term variations in direction (Bristow et al., 2000; Beveridge et al., 2006; Sridhar et al., 2006; Bristow et al., 2007; Zhou et al., 2012; Telfer and Hesse, 2013; Liu and Baas, 2020), although dune morphology also depends on sediment availability, erodibility, and
vegetation covers (du Pont et al., 2014; Gao et al., 2015). The spatial distribution of palaeo-wind patterns recorded in aeolian dune slip-face azimuths thus allow the deduction of atmospheric circulation patterns in geological periods characterised by different land–sea distributions (Parrish and Peterson, 1988; Peterson, 1988; Scherer, 2000; Loope et al., 2004) or different palaeoclimatic settings (Beveridge et al., 2006; Sridhar et al., 2006; Rodríguez-López et al., 2008; Hasegawa et al., 2012).

Peterson (1988) initially described the spatial and temporal changes in palaeo-wind regimes from Carboniferous to Middle Jurassic aeolian sandstone in the western United States (US) (i.e., the Colorado Plateau and surrounding area). Using these datasets and published paleomagnetic data, Loope et al. (2004) suggested the predominance of a desert environment in the Pangaean equatorial area (paleolatitude 8°–13°N), with a broad sweep of SW-ward winds in the northern area curving to SE-ward in the southern area during the Early Jurassic.
However, the reconstructed surface wind pattern shows discrepancies with the results of model-based reconstructions (Rowe et al., 2007), even though the revised paleolatitude of the Colorado Plateau (17°–24°N) has been used. Rowe et al. (2007) suggested several possible reasons for this discrepancy, including (1) the palaeomagnetism-based palaeogeographic reconstructions of the Jurassic are incorrect, (2) the interpretation of how winds shaped the dunes is mistaken, or (3) the basic climate controls during the Jurassic were different from
those of today.

We considered the problems involved in the measurement of palaeo-wind direction data from the aeolian dune record. Peterson (1988) provided only single preferred palaeo-wind directions at each site without information on dune morphology, thus hindering the understanding of accurate palaeo-wind flow regimes. Re-evaluation of palaeo-wind direction data with a particular focus on dune morphology and multi-directional wind
regime therefore could provide an explanation for the discrepancy between reconstructed surface wind patterns and model-based reconstructions. In addition, recent palaeomagnetic studies that addressed the inclination-shallowing problem (Kent and Irving, 2010; Dickinson, 2018) provide the revised palaeolatitude of the Colorado



Plateau as N19°–27° during the Early Jurassic, which corresponds to the location of the model-generated desert area and the subtropical high-pressure belt (Rowe et al., 2007). To solve the discrepancies between model experiments and the reconstruction from aeolian dune records, re-evaluation of the Lower Jurassic aeolian dune slip-face azimuth record considering the latest palaeogeographical reconstruction is required.

Here, we present the measurement results of spatial distribution of slip-face azimuths and inferred dune morphology recorded in Lower Jurassic aeolian sandstones in the western US, together with the latest palaeolatitude data (Kent and Irving, 2010; Dickinson, 2018). Our measurement of slip-face orientations over a wide area of the western US and field observational evidence reveals that the presence of longitudinal dunes in the Early Jurassic desert that formed as a result of tri-directional palaeo-wind patterns, consistent with model-generated seasonal wind regimes (Rowe et al., 2007). We also discuss the formation process of longitudinal dunes in relation to orbitally induced changes in atmospheric pressure configuration, by comparing climate model reconstruction in Pangaea supercontinent (Winguth and Winguth, 2013) and geological evidence for late Quaternary dune alignments (Lancaster, 1981; Lancaster, 1990; Bristow et al., 2000; Lancaster et al., 2002; Bristow et al., 2007).

## 2 Material and Methods

### 2.1 Lower Jurassic aeolian sandstone

To obtain the spatial distributions of palaeo-wind directions, we surveyed Lower Jurassic aeolian sandstone strata and measured aeolian dune maximum slip-face azimuths over a wide area, in the western US; specifically, the Navajo Sandstone on the Colorado Plateau (Utah, Colorado, and Arizona) and correlative strata of the Nugget Sandstone to the north (Idaho, Wyoming) and the Aztec Sandstone to the south (Nevada) (**Fig. 1**). The maximum thickness of the strata is approximately 700 m, in southern-central Utah, and thin to approximately 100–150 m toward the northern area (Nugget Sandstone in Idaho and Wyoming) and eastern area (Glen Canyon Sandstone in western Utah and Colorado) (Blakey et al., 1988; Parrish and Peterson, 1988; Peterson, 1988; Blakey, 2008). This palaeo-dune field covers a vast area of approximately 625,000 km$^2$, an area 2.5 times larger than surface and subsurface extent of the strata (Tape, 2005). On the basis of existing palaeomagnetic studies (Kent and Irving, 2010; Dickinson, 2018), the palaeolatitude of the studied palaeo-dune field is inferred to have been ca. 19°–27°N during the Early Jurassic, with clockwise continental rotation of ~5°.

Chronology and correlation of the Lower Jurassic aeolian sandstone (Navajo Sandstone and correlative strata) in western US is currently debated (Dickinson and Gehrels, 2009; Dickinson et al., 2010; Sprinkel et al., 2011; Rowland and Mercadante, 2014; Parrish et al., 2019). Based on the U–Pb age dating of detrital zircons, the depositional ages of the underlying Kayenta Formation and the overlying Page Sandstone are considered to be 190–187 Ma and 170 ±3 Ma, respectively (Dickinson and Gehrels, 2009). From the U-Pb and $^{40}$Ar/$^{39}$Ar geochronology for pyroclastic zircon and biotite crystals from tephra lenses in the basal part of the Page Sandstone further provide the upper limit age of the Navajo Sandstone as 172.3–170.6 Ma (Dickinson et al.,



2010). Based on these existing chronological data-sets, the duration for deposition of the Lower Jurassic aeolian sandstone is estimated as ca. 14.7–19.4 Myr. On the other hand, Parrish et al. (2019) recently proposed significantly older age (200.5 ±1.5 Ma and 195.0 ±7.7 M) for the Navajo Sandstones in southeastern Utah based on the U-Pb analyses of carbonate deposits. This new chronological data suggests that the basal part of the Navajo Sandstone and the underlying Kayenta Formation are interfingering and/or even time-transgressive over a few million years (Parrish et al., 2019).

Stratigraphic correlation of cross-strata in the Lower Jurassic aeolian sandstone over a wide area of western US is challenging due to the chronological problems described above and lack of comparable first-order bounding surfaces. Although it is difficult to accurately correlate the strata in each regions, we investigated the records over a wide area in order to obtain spatial distribution pattern of palaeo-wind directions as shown in **Fig. 1**. The obtained results may not represent a truly contemporaneous data-sets, and the exact spatial distribution pattern in the same time-window needs further chronological research. Nevertheless, we believe that the spatial distribution of palaeo-wind directions presented in this study is important because it represent integrated pattern of mesoscale wind regimes in western US area of Pangaea supercontinent during the Early Jurassic.

## 2.2 Reconstruction of spatial palaeo-wind patterns

Palaeo-wind directions at each sites were determined based on the measurement of maximum slip-face dip azimuths at each outcrops of cross-bedding strata. We measured totally 1636 slip-face azimuth data from 178 sites (**Supplementary table**). Unlike the Peterson (1988), which demonstrated only single preferred palaeo-wind directions at each site, we provided multi-directional palaeo-wind regime as rose diagrams shown in **Fig. 1**. We also showed regional-scale slip-face azimuth data and outcrop evidence of the Navajo Sandstone in Zion National Park, southwestern Utah, and Arches National Park, eastern Utah (**Figs. 2, 3**). The obtained slip-face azimuth data were corrected for the magnetic declination (+11°) in the western US (Thébault et al., 2015) and bedding tilts, using the Kyoto Untilting Tool software (Tomita and Yamaji, 2003) developed by the Yamaji Laboratory, Kyoto University. The obtained slip-face azimuth data were then corrected for post-Jurassic continental clockwise rotation (5°) and plotted on a paleogeographic map (**Fig. 1**) with paleolatitudes based on paleomagnetic data (Dickinson, 2018; Kent and Irving, 2010). The obtained palaeo-wind data were plotted on rose diagrams using the analytical software "Rose" developed by the Naruse Laboratory, Kyoto University.

The obtained slip-face azimuth data commonly shows multiple direction patterns. To statistically separate multiple components and calculate preferred median directions, we use "Gaussian Mixture (GM) model" assuming that data distribution is generated from a mixture of a finite number of Gaussian (Normal) distribution. Using Expectation-maximization (EM) algorithm fitting the GM model, we separated slip-face azimuth data into multiple components (distributions), and finally calculated median palaeo-wind directions at each site. Component numbers are estimated either by clustering analysis or numbers of major peaks in histograms of slip-face azimuth data at each sites (**Table 1**).



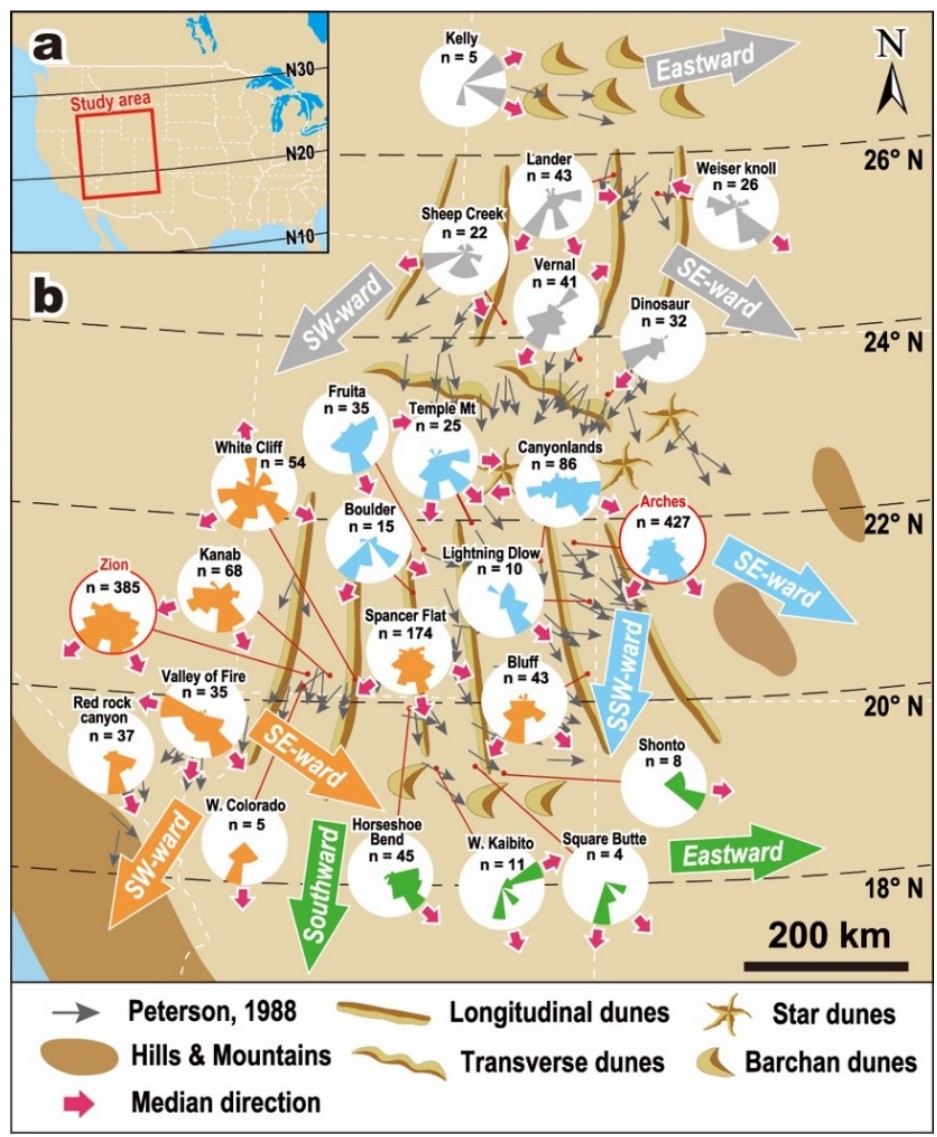

**Figure 1: Reconstructed palaeo-wind patterns in the western US during the Early Jurassic. (a) Location of the study area in the western US. (b) Spatial distributions of dune slip-face azimuths measured in this study (shown on rose diagrams) and inferred dune morphologies, plotted on the Early Jurassic palaeogeographic map of Peterson (1988) with palaeolatitudes based on Dickinson (2018). Rose diagrams in each areas are shown by different colour: northern (grey), central (light blue), southwestern (orange) and southern (green) parts of study area. Large-size arrows shown in (b) indicate reconstructed prevailing surface wind directions in each area. The pink coloured arrows in each rose diagrams indicate median directions of separated components in slip-face azimuth data shown in Table 1.**





**Table 1**: **Calculated median directions of slip-face azimuths in each sites. Original data is shown in Supplementary Table 1.**

| Study sites | Number of data | Median direction 1 | Median direction 2 | Median direction 3 | Study sites | Number of data | Median direction 1 | Median direction 2 | Median direction 3 |
|---|---|---|---|---|---|---|---|---|---|
| Kelly* | 5 | 63.5° | 116.0° | – | Zion | 385 | 156.3° | 228.3° | – |
| Lander* | 43 | 91.0° | 156.0° | 212.0° | Kanab | 68 | 156.4° | 252.1° | – |
| Weiser knoll | 26 | 129.7° | 290.5° | – | White Cliff | 54 | 116.2° | 236.2° | 351.7° |
| Sheep Creek | 22 | 163.8° | 260.6° | – | Spencer Flat | 174 | 111.3° | 166.6° | 230.4° |
| Vernal | 41 | 49.7 | 212.7° | – | Bluff* | 43 | 135.5° | 211.0° | – |
| Dinosaur | 32 | 223.9° | – | – | Red rock canyon | 37 | 159.3° | – | – |
| Fruita | 35 | 81.5° | 159.4° | – | Valley of Fire | 35 | 143.3° | 192.0° | 282.8° |
| Temple Mountain* | 25 | 94.0° | 135.0° | 185.0° | Western Colorado | 5 | 182.0° | – | – |
| Canyonlands | 86 | 112.3° | 264.0° | – | Horseshoe Bend | 45 | 134.6° | – | – |
| Arches* | 427 | 146.0° | 211.0° | – | Western Kaibito | 11 | 64.0° | 165.2° | – |
| Lightning Dlow | 10 | 132.0° | – | – | Square Butte | 4 | 137.6° | 188.5° | – |
| Boulder | 15 | 118.3° | 202.8° | – | Shonto | 8 | 96.0° | – | – |

*Separation of components and median directions in these sites are calculated based on histogram of data distribution.

## 3 Results and Discussion

### 3.1 Development of NNE–SSW- to NNW–SSE-oriented longitudinal dunes

The spatial distribution of dune slip-face azimuths obtained in this study indicates multiple directions of palaeo-wind flow, with overall tri-modal preferred directions of eastward, southeastward, and southwestward (**Fig. 1b**). Although a previous study (Peterson, 1988) demonstrated a single preferred palaeo-wind direction for each region, which is overall consistent with our data, the obtained datasets show marked multiple directional patterns. In addition, slip-face azimuths in most of the regions show bi-directional and oblique angular variation between ~80° and 135°.

Outcrop evidence of slip-face azimuths of the Navajo Sandstone in Zion National Park and Arches National Park are illustrated in **Figs. 2 and 3**, respectively. The rose diagrams for Zion region show bi-modal preferred directions to the SSE and SW (centred at ~160° and ~240°). In addition, the preferred direction switches between SSE-wards and SW-wards at intervals of approximately 1 km (**Fig. 2a**). The rose diagrams for Arches region also show bi-modal preferred directions to the SE and SSW (centred at ~140° and ~220°) with switching of preferred directions at intervals of approximately 1–2 km (**Fig. 3a**). The outcrop, which is located in the boundary of bi-directional and oblique angular slip-face directions, shows zigzagging patterns and compound sets of cross-stratification in both regions (**Figs. 2c, 3c**). The observed bi-directional cross-bed structures exhibit a marked





correspondence to the internal structures of modern longitudinal dunes as reconstructed on the basis of ground-penetrating radar (GPR) profiles (Bristow et al., 2000; Bristow et al., 2007; Zhou et al., 2012; Telfer and Hesse, 2013; Liu and Baas, 2020) (**Appendix A**). Internally, modern longitudinal dunes exhibit oblique bi-directional cross-beds on each side of the dune flank and cross-beds in both directions in the central part. Optically

stimulated luminescence (OSL) age dating has further elucidated the timing and duration of cross-beds and the wind regimes during formation of cross-bedding (Bristow et al., 2000; Bristow et al., 2007; Zhou et al., 2012). Sets of trough cross-stratification are formed in the central dune crest by superposition of bi-modal dunes. The change in preferred direction at 1–2 km intervals (**Figs. 2a, 3a**) is also consistent with the spacing of modern longitudinal dunes (Wasson and Hyde, 1983; Lancaster, 2006).


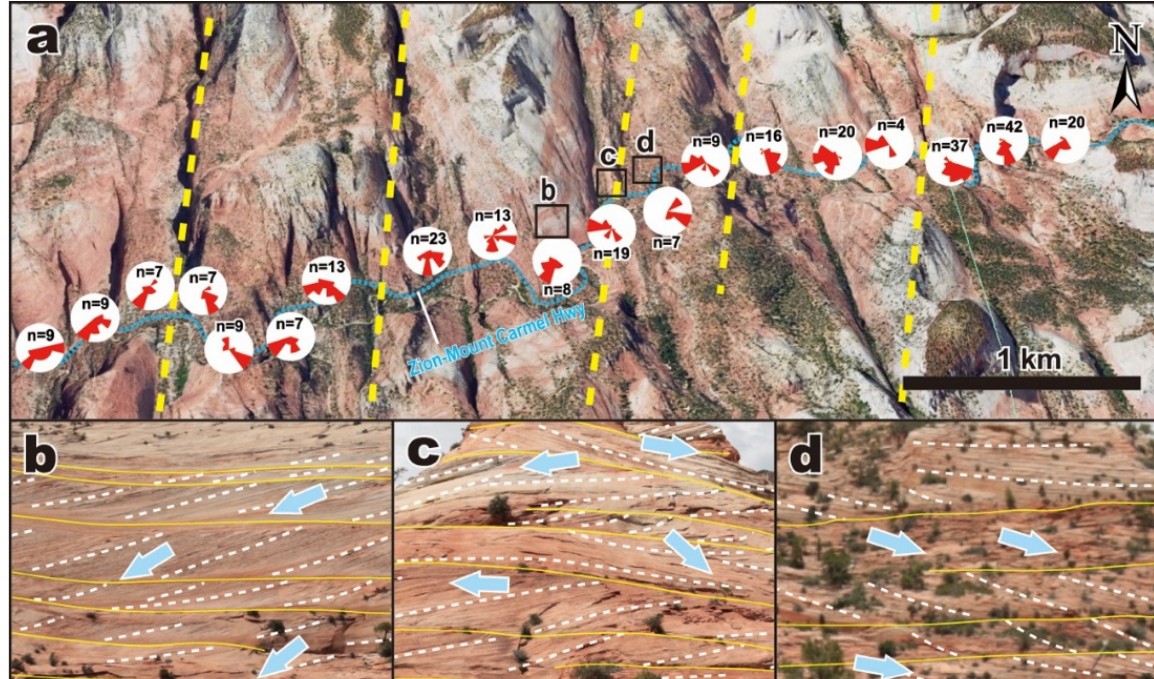

**Figure 2: Spatial pattern of dune slip-face azimuths and outcrop photographs of the Navajo Sandstone in the Zion National Park. (a) Spatial distribution of palaeo-wind data shown by rose diagrams along the Zion–Mount Carmel Highway. Yellow dotted lines are the inferred locations of the central crests of longitudinal dunes, which are spaced**
**at approximately 1 km intervals. (b–d) Outcrop photographs of cross-stratification structures in the Navajo Sandstone. Yellow solid lines and white dashed lines indicate bounding surface of aeolian dune strata and slip-face cross-stratification, respectively. All photographs taken facing to north.**



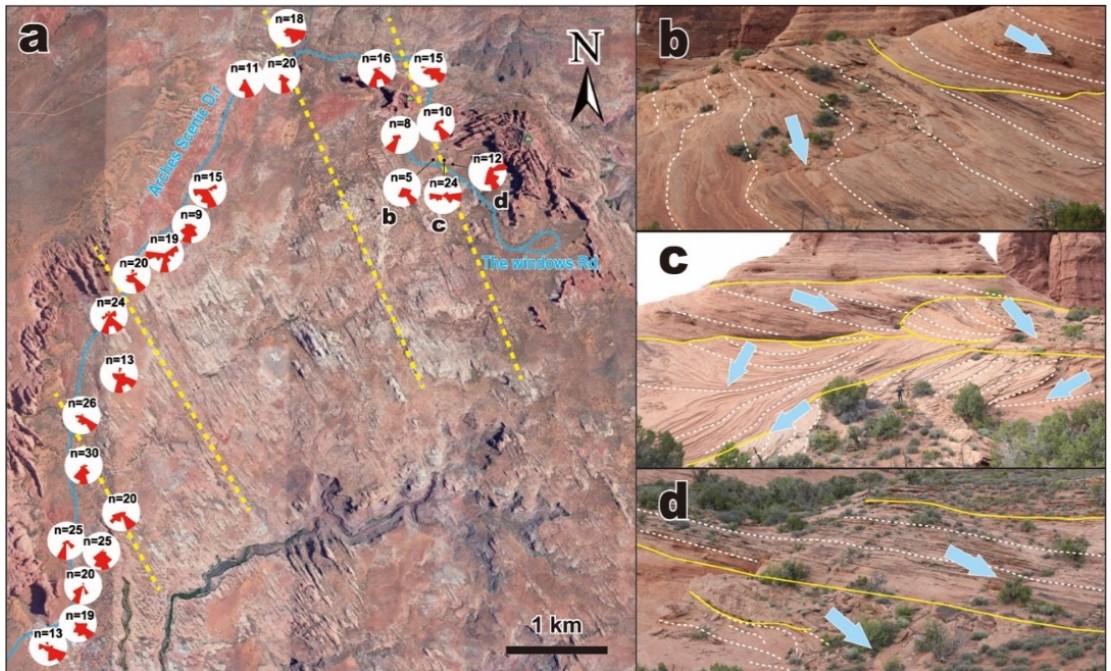

**Figure 3: Spatial pattern of dune slip-face azimuths and outcrop photographs of the Navajo Sandstone in the Arches**
**National Park. (a) Spatial distribution of palaeo-wind data shown by rose diagrams along the Arches Scenic Drive.**
**Yellow dotted lines are the inferred locations of the central crests of longitudinal dunes. (b–d) Outcrop photographs**
**of cross-stratification structures in the Navajo Sandstone. Yellow solid lines and white dashed lines indicate bounding**
**surface and slip-face cross-stratification, respectively. All photographs taken facing to north.**

The angle of the observed bi-modal directions in the slip-face azimuth exhibits good agreement with
experimental estimates of the angle of flow direction to form longitudinal dunes. Water flume experiments
simulating the formation of different types of dune have suggested that longitudinal dunes form under bi-
directional flows with angles of 90°–135° between flows, and lie along the average wind direction (Taniguchi et
al., 2012). The formation of longitudinal dunes by seasonal turnover of oblique wind flow is also consistent with
numerical modelling (Parteli et al., 2014; Gao et al., 2015; Liu and Baas, 2020) and observational studies (Breed
et al., 1979; Wasson et al., 1988; Livingstone, 1989; Hesse, 2010; Zhou et al., 2012). It should be noted that
longitudinal dunes are also forms under uni-modal wind regime by the influence of vegetation, clay and salt
content, and related sediment cohesiveness, whilst sinuous uni-directional dunes are also forms under bi-modal
wind regimes by the influence of sand availability (du Pont et al., 2014; Gao et al., 2015). However, zigzagging
patterns and compound sets of cross-stratification preserved in Navajo Sandstone of Zion and Arches regions
(**Figs. 2c, 3c**) are resembling to inferred longitudinal dune of other aeolian dune strata (Scherer, 2000; Abrantes
et al., 2020). In addition, GPR observation of modern sinuous uni-directional dune (Fu et al., 2019) indicate bi-
directional but rather thinner cross-bed structures compared with that of longitudinal dunes (Bristow et al., 2000;
Bristow et al.,2007; Zhou et al., 2012; Telfer and Hesse, 2013; Liu and Baas, 2020). Therefore, Navajo Sandstone



in Zion and Arches regions are interpreted to be formed by longitudinal dunes, consistent with earlier suggestion
by Rubin and Hunter (1985).

In addition to these outcrop evidence in Zion and Arches regions, the spatial pattern of slip-face directions suggests that longitudinal dunes are widely distributed in the western US that were oriented NNW–SSE to NNE–SSW in the palaeolatitude range ~20°–26°N during the Early Jurassic (**Fig. 1b**). The obtained data further represent characteristic wind flow regimes in each of the four areas. Specifically, the slip-face azimuths in the
southernmost area (palaeolatitude: ~19°N) show a bi-directional pattern of eastward and southward palaeo-winds, whereas the southwestern and southeastern areas (palaeolatitude: ~20°–22°N) show preferred SW-ward and SE-ward directions, with an increased influence of westward palaeo-winds in the southwestern area. The central area (palaeolatitude: ~22°N) shows more complex and multi-directional patterns, suggesting the presence of some star dunes (Lancaster, 1989). In contrast, the southern part of northern area (palaeolatitude: ~23°N) shows uni-
directional preferred SW-ward directions suggesting the dominance of transverse dunes, which is also consistent with the evidence of NW–SE-oriented underground dune textures reconstructed by means of seismic inversion (Verma et al., 2018). The northern area (palaeolatitude: ~24°–26°N) shows bi-directional palaeo-wind pattern of SW-ward and SE-ward directions, whereas the northernmost area (palaeolatitude: ~27°N) shows more influence of eastward palaeo-wind flows.

The observed tri-directional pattern of slip-face azimuths suggests that the longitudinal dunes were formed as the result of a combination of westerly, northwesterly, and northeasterly palaeo-winds (**Fig. 1b**). The reconstructed tri-modal paleo-wind directions exhibit good correspondence with model-generated prevailing surface wind patterns, such as the northwesterly winds developed over the study area during the boreal summer, whereas northeasterly trade winds (ca. 20°–25°N) and westerlies (ca. 25°–30°N) developed during the boreal
winter (Rowe et al., 2007).

### 3.2 Orbital-scale changes in Pangaean atmospheric pressure configuration

Observed bi-modal slip-face azimuth patterns and outcrop evidences also implies influence of orbital-scale changes in atmospheric pressure regimes. The vertical profile of inferred central crests of longitudinal dune strata in Zion and Arches shows trough-shaped cross-stratified structures with SW-ward and SE-ward slip-face
azimuths (**Figs. 2c, 3c**). The inversion of this paleo-wind pattern appears to occur every ca. 2–3 m of stratigraphic thickness. Based on the existing chronological data (Dickinson and Gehrels, 2009; Dickinson et al., 2010; Parrish et al., 2019), we assume the duration of deposition of the Lower Jurassic aeolian sandstone to be ca. 14.7–19.4 Myr with a thickness of 300–700 m; thus, the average accumulation rate of the deposits can be calculated to be ca. 1.5–4.8 cm/kyr. Based on this estimated accumulation rate, paleo-wind inversion (every 2–
3 m) occurred every 42–200 kyr of the orbital-change time-scale (i.e., 20-kyr precession, 40-kyr obliquity and 100-kyr eccentricity cycles). This interpretation is consistent with the evidence of fluvial–aeolian depositional cycle of Navajo–Kayenta transition in Kanab, Utah (~20 m) (Hassan et al., 2018) and Permian aeolian–alluvial



depositional cycles in Colorado (2–15 m cycles) (Pike and Sweet, 2018), which are both interpreted to reflect the 100- and 400-kyr eccentricity cycle. Although GPR observation and OSL dating of modern longitudinal dune formation (Bristow et al., 2000; Bristow et al., 2007) have revealed that the time-scale of dune cross-bed is slightly shorter (centennial- to millennial-scale), orbital-scale changes in the area of the dune formation field likely cause much a longer time-scale of vertical dune accumulation.

Using observed multi-directional dune slip-face azimuth patterns (**Fig. 1b**), in conjunction with the model simulation result (Winguth and Winguth, 2013), we illustrated the inferred seasonal- and orbital-scale changes in wind regime and the resulting dune alignments for precession maximum and minimum (**Fig. 4**). The principal difference resulting from orbitally induced changes in atmospheric pressure configuration is the predominance of a low-pressure system (continental) at ~20°–25°N during the boreal summer at precession maximum (Winguth and Winguth, 2013). The resulting large low-pressure cell diverts the moist tropical air masses of the ITCZ away from Panthalassa (**Fig. 4c**). Intense summer rainfall in the southern area (~20°–22°N) and resulted vegetation and soil covers likely stabilized dune-sand movement and resulted in enhanced sand accumulation ( Lancaster and Baas, 1998; Hesse and Simpson, 2006; Kocurek and Ewing, 2012), consistent with the maximum thickness of the Navajo Sandstone in south-central Utah (Blakey et al., 1988; Parrish et al., 2019). Increased humidity and vegetation in the southern area is also consistent with the evidence of nodular layers and trace fossils within dune slip-faces in south-central Utah (Chan and Archer, 2000; Loope et al., 2001; Ekdale et al., 2007). In the northern areas, relatively dry NNW winds prevailed during summer (**Fig. 4c**). In contrast, owing to the development of the high-pressure system at ~30°N, dry NE trade winds prevailed over study area during winter (**Fig. 4a**). Because of this seasonal turnover of bi-modal winds, NNE–SSW- to N–S-oriented longitudinal dunes were developed during precession maximum (**Fig. 4a, c**).

In contrast, owing to the development of a high-pressure system, arid climatic conditions prevailed in the whole study area in both summer and winter during precession minimum (Winguth and Winguth, 2013) (**Fig. 4b, d**). In the central and southern areas, winter NE trade winds and summer NNW to NW winds occurred; as a result, NNW–SSE-oriented longitudinal dunes were developed in these areas (~20°–22°N). In the northern area at around 23°N, star dunes likely formed as a result of multi-directional wind flows in the center of a high-pressure system in winter. In the northernmost area (~27°N), westerly winds dominated in both summer and winter, so eastward-moving barchanoid dunes likely developed there during precession minimum (**Fig. 4b**).



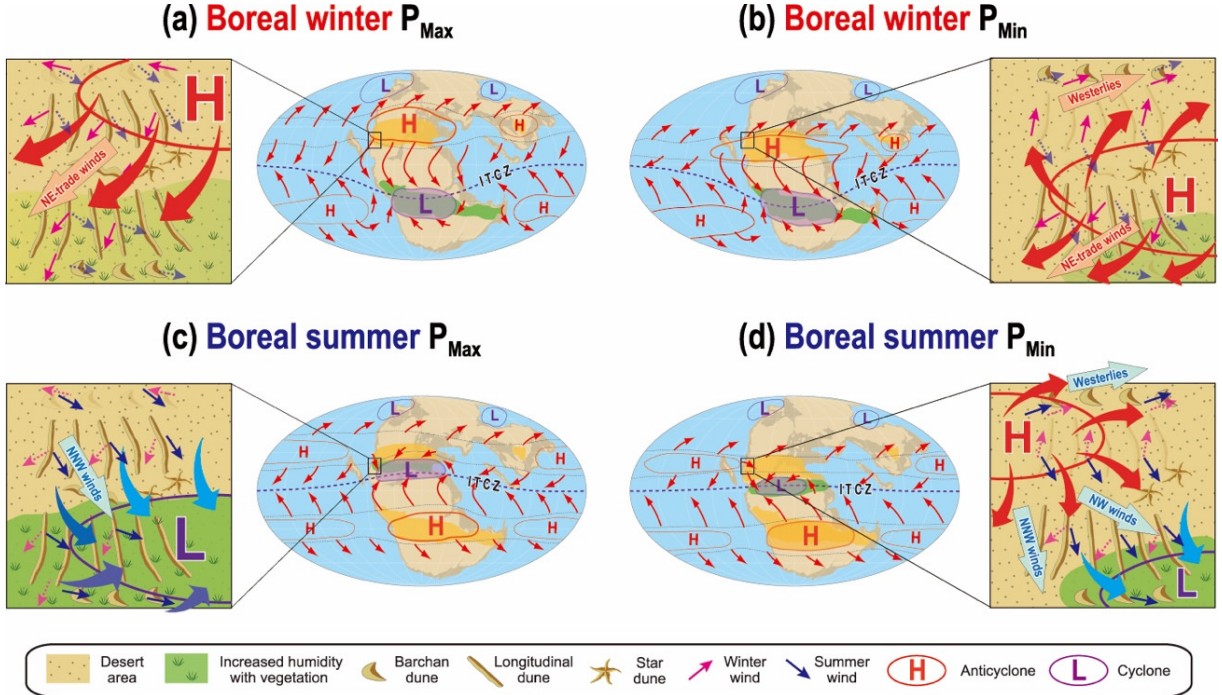

**Figure 4: Schematic illustrations of orbital and seasonal changes in Pangaean atmospheric pressure configuration and inferred dune alignments in western US during the precession maximum ($P_{Max}$; a, c) and precession minimum ($P_{Min}$; b, d). Orbital-scale changes in pressure configuration are illustrated based on climatic model results (Winguth and Winguth, 2013). Seasonal changes in wind regimes are indicated as pinkish (boreal winter) and bluish (boreal summer) arrows. During the precession maximum, continental low-pressure system in boreal summer likely reduced by a few hPa relative to the precession minimum in response to the surface temperature rise (Winguth and Winguth, 2013). Increased moisture and vegetation cover likely resulted in dune stabilization in the southern area during the boreal summer at precession maximum (c).**

Orbital-scale changes in atmospheric pressure configuration and the relationship of these changes to dune formation in Pangaea are supported by evidence from late Quaternary dune fields in South Africa (Lancaster, 1981; Thomas and Burrough, 2016; Thomas and Bailey, 2017). Luminescence dating of dunes in the Kalahari Desert has revealed that the timing of dune accumulation varies broadly from the northeastern Kalahari (~16°–20°N; ~35 ka and later part of the Last Glacial Maximum) to the southwestern Kalahari (~23°–30°N; ~13.5 ka and Holocene) (Thomas and Burrough, 2016; Thomas and Bailey, 2017). In addition, increased dune accumulation appears to be coincided with immediately after summer insolation maximum. A geomorphological study (Lancaster, 1981) has also suggested that the pattern of dune alignment in the Kalahari Desert reflects changes in atmospheric pressure regime between the Holocene and the last glacial (Stone, 2014) (**Appendix B**).

Evidence for the accumulation ages of late Quaternary dune fields in Sahara Desert (Lancaster et al., 2002; Bristow and Armitage, 2016) also has important implications for the relationship between dune stabilization and climatic humidity. Increased summer rainfall resulting from a northward shift of the ITCZ in the southern Sahara during the Early Holocene led to the so-called "Green Sahara" (Kuper and Kröpalin, 2006; Pausata et al., 2016); however, luminescence age dating of dune sands in the southern Sahara has revealed increased dune



accumulation during this period, suggesting that dune stabilization proceeded in moist conditions (Bristow and Armitage, 2016). This observation is consistent with our interpretation that intense summer rainfall with vegetation and soil covers during the precession maximum likely stabilized dune-sand accumulation (Lancaster and Baas, 1998; Hesse and Simpson, 2006; Kocurek and Ewing, 2012) and resulted in the development of nodular layers and trace fossils within the Navajo Sandstone strata in the southern Utah (Chan and Archer, 2000;

Loope et al., 2001; Ekdale et al., 2007). Stabilized dune sand accumulation by vegetation and soil cover likely lead to the notably thick Navajo Sandstone succession (Blakey et al., 1988; Parrish et al., 2019) in southwestern area of Colorado Plateau.

## 4 Conclusion

In summary, by comparison with climate model reconstruction and geological evidence of late Quaternary dune
fields, reconstructed Lower Jurassic longitudinal dunes in the western US likely reflect seasonal- and orbital-scale changes in the wind regime and atmospheric pressure configuration over Pangaea. Desert development and predominant westerly winds in equatorial Pangaea during the Early Jurassic, which is suggested by Loope et al. (2004), are not supported by revised palaeomagnetic data and the palaeo-wind regimes obtained in this study. NNE–SSW- to NNW–SSE-oriented longitudinal dunes in the central and southern area, with eastward-migrating

uni-directional barchanoid dunes in the northernmost and southernmost parts, are interpreted to have formed as the result of superimposed integration of seasonal and orbital changes in wind regimes. The reconstructed palaeo-wind pattern at ~19°–27°N appears to be consistent with model-generated surface wind patterns and the location of the subtropical high-pressure belt. Therefore, we have solved the enigma of Pangaean atmospheric circulation patterns, such as discrepancy between model-generated wind directions and aeolian dune records noted by Rowe

et al. (2007). The thick Navajo Sandstone succession in the southern area likely reflects increased moisture and resulting dune stabilization by the northward shift of the ITCZ during the precession maximum, which is consistent with evidence of dune accumulation in the "Green Sahara" during the Early Holocene.

## Appendix A

### Comparison between the internal structures of modern longitudinal dunes and Lower Jurassic aeolian
**cross-strata**

Outcrop evidences of the Lower Jurassic aeolian cross-sets exhibit a marked correspondence to the internal structures of modern longitudinal dunes. Internally, modern longitudinal dunes exhibit oblique bi-directional cross-beds on each side of the dune flank and stacking of cross-beds in both directions in the central part. In central part of longitudinal dune, vertical stacking of two oblique opposing directional cross-sets result in

increasing of dune height (Bristow et al., 2000; Bristow et al., 2007; Zhou et al., 2012; Telfer and Hesse, 2013; Liu and Baas, 2020).



These structural feature of modern longitudinal dunes are also observed in some outcrops in Lower Jurassic aeolian strata in western US. Some outcrops in the Zion National Park and the Arches National Park, which are located in the boundary of bi-directional slip-face directions (**Figs. 2, 3**), show zigzagging patterns and compound

sets of cross-stratification (**Fig. A1**).  Both outcrops exhibit vertical stacking of two oblique directional cross-beds. Thus, sedimentary structures in the Zion and Arches regions are interpreted to have been formed by vertical stacking cross-sets of longitudinal dunes.

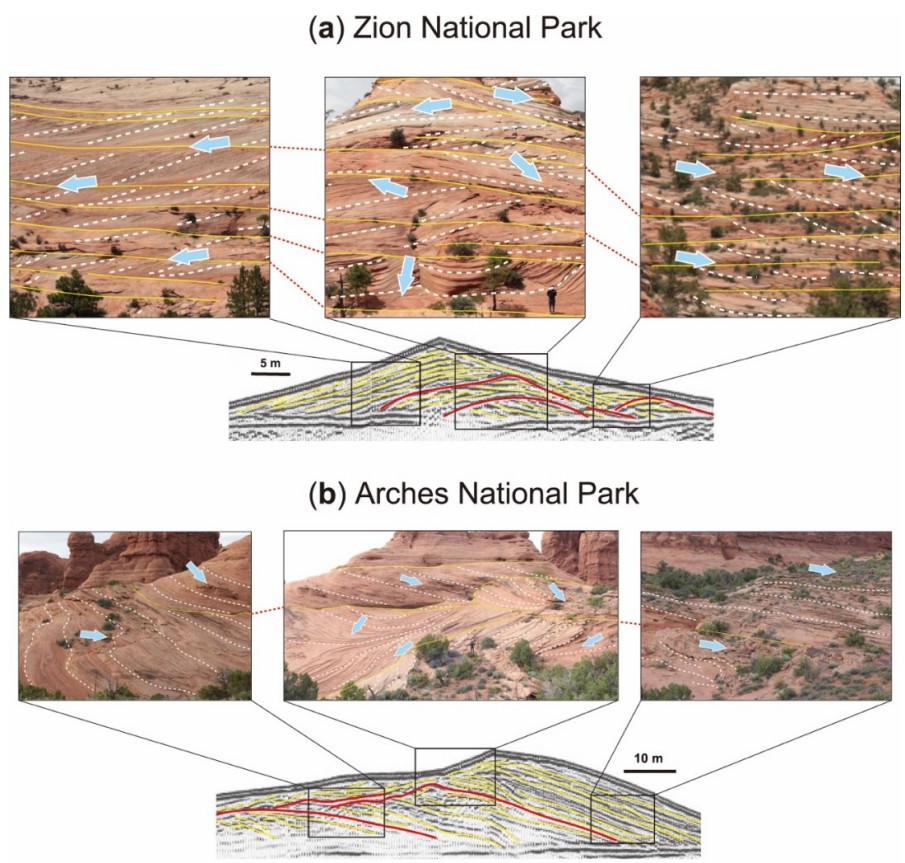

**Figure A1: Comparison between the internal structures of modern longitudinal dunes (Bristow et al., 2000) and the outcrop photographs of Lower Jurassic aeolian cross-stratifications exposed in Zion National Park (a) and Arches National Park (b). Yellow solid lines and white dashed lines indicate bounding surface of aeolian dune strata and slip-face cross-stratification, respectively. Yellow and red traces in the ground-penetrating radar (GPR) profiles of modern longitudinal dunes (Bristow et al., 2000) indicate slip-faces and unconformities, respectively.**

**Appendix B**

**Late Quaternary dune alignment and orbital changes in wind regime in South African desert**

Orbitally induced changes in atmospheric pressure configuration and resulted changes in dune alignment at ~19°–27°N of Pangaea Supercontinent are supported by evidence from late Quaternary dune records in South





Africa (Lancaster, 1981; Thomas and Burrough, 2016; Thomas and Bailey, 2017). Geomorphological study
suggested that the NNW–SSE-oriented longitudinal dunes in the southern and western Kalahari likely formed
by the interactions of austral winter northeasterlies from the continental high and summer westerlies from the
South Atlantic Anticyclone (SAA) during the Holocene (Lancaster, 1981). Dune fields in the northern and
eastern Kalahari were covered by vegetation as a result of increased rainfall caused by the dominance of a low-
pressure cell in inland South Africa (Lancaster, 1981; Stone, 2014). In contrast, during the last glacial period,
the northern and eastern Kalahari dune fields experienced enhanced aridity as a result of an equatorward shift of
the continental high, whereas the southern and western Kalahari dune fields became more humid as a result of
moist westerly winds from the SAA (Stone, 2014) (**Fig. B1**). These evidence of orbital-scale changes in wind
regime and dune development area during the Holocene and the last glacial can provide significant implication
for understanding of such changes in Pangea Supercontinent during the Early Jurassic period.


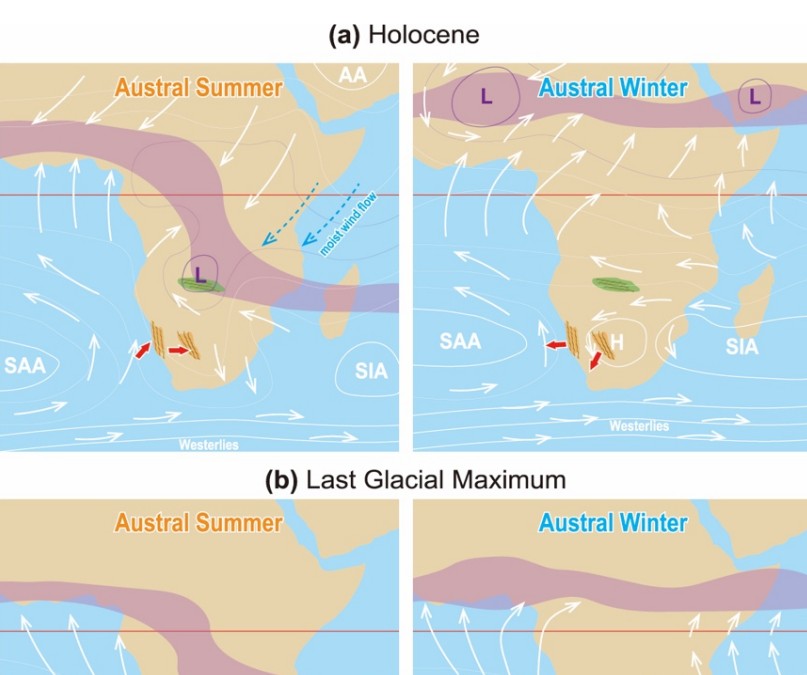

**Figure B1: Schematic illustrations of changes in atmospheric pressure configuration and surface wind patterns in
South Africa during the Holocene (a) and the Last Glacial Maximum (b). The purplish shaded area marks the
intertropical convergence zone. White arrows indicate dominant surface winds (modified after, Stone, 2014). Bold red
arrows indicate inferred wind flows, which formed longitudinal dunes. Yellowish and greenish area indicates dry and
moist dune system of northern and southern Kalahari and Namib deserts, respectively. H: High pressure, L: Low
pressure, SAA: South Atlantic Anticyclone, SIA: South Indian Anticyclone, AA: Arabian Anticyclone.**



**Data availability**

All data-sets are shown in the main text, appendix and supplementary material.

**Author contributions**

H.H. designed this research. H.S. and H.H. conducted field survey and wrote the manuscript.

**Acknowledgements**

We greatly appreciate H. Asahi for his assistance of statistical analysis of palaeo-wind direction data using EM algorithm by Matlab software. We also thank M. Ikeda and R. Kuma for discussions and field assistance.

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
