# Peer review of "Development of longitudinal dunes under Pangaean atmospheric circulation"

_Climate of the Past, 2021_

## Referee Comment (RC3)

[referee-annotated manuscript omitted]

---

## Author Response (AR1)

**Responses to comments from Anonymous Referee #1**

We are grateful to the referee #1 for the constructive comments and suggestions. Below, we give our responses in turn following each comment, with the reviewers' comments being in with underline and our responses being in without underline.

1. A major comment is the teminology of atmospheric circulations. I would suggest authors use monsoonal circulation, instead of the Hadley circulation. The former is regional, while the latter is zonal-mean circulation. Please read Parrish (1993).

According to the referee's suggestion, we revised the term "cross-equatorial Hadley circulation" to "cross-equatorial flow induced by monsoonal circulation" in Line 11 and 27-28 in the revised manuscript.

2. In some places, "palaeo" is used, while "paleo" is used in other places. It is better to integrate them.

We unified into "palaeo" in the revised manuscript.

3. Line 227: "evidences" --> evdience,

We have corrected it as suggested.

4. Line 302: the equatorial Pangea

We have corrected it as suggested.

5. Line 316: evidence .. exhibits

We have corrected it as suggested.

**Responses to comments from Anonymous Referee #2**

We are grateful to the referee #2 for the constructive comments and suggestions. Below, we give our responses in turn following each comment, with the reviewers' comments being in with underline and our responses being in without underline.

In large part the paper is well-written and structured, the evidence appears sound and the main conclusions seem to me to be well founded.
The aspects of the paper which deal with palaeo-wind directions, establishment of the aeolian sands as being deposited by longitudinal dunes, and the palaeogeographic interpretation I thought were unproblematic.
However, the paper also made some highly speculative forays into the periodicity of past aeolian events, timing and association with orbital forcing and then to reconciliation with orbitally-forced climate simulations which I thought were unsupported. Principally, the authors did not recognize that any sediment sequence (and perhaps particularly sand dunes) is incomplete. They suffer erosion almost synchronously with deposition and for long periods afterwards until deeply buried. This is quite well modelled in papers cited by the authors (Thomas and Bailey, 2017). With only very broad dating available (millions of years) it is impossible to establish if the preserved cross-bed sets had any temporal or causal association with orbital cycles. Some description of the bounding surfaces may have gone part way to answering this, but see Leighton et al, 2013, QSR; 2014 ESR, for cautionary tales. I think that this section, specifically any claims for orbital forcing, should be removed from the paper.

In response to referee's comments, we extensively revised the discussion of orbital cycles. We recognize that Quaternary evidence does not support a clear relationship between dune-field activity and orbital forcing (e.g., Thomas and Bailey, 2017); thus, we deleted almost all of the sentences related to orbital-scale changes of dune-fields based on Quaternary records, as well as **Appendix B** and **Fig. B1**. Instead, we focus on comparison of the observed surface wind pattern with model results. As part of this change in focus, we altered the title of section 3.2 to "Comparison of modelled and observed surface wind patterns".

In the last part of section 3.2, we retained some discussion of the possibility of orbital-scale climatic change being recorded in aeolian depositional sequences in subtropical Pangaea, using evidence from previous studies. For instance, we added the following sentences: "Previous studies also raised the possibility that orbital-scale climatic changes recorded in fluvial–aeolian cycles (~20 m thick) in the Lower Jurassic Navajo–Kayenta transition in Utah (Hassan et al., 2018), aeolian cyclic sequences (~4–8 m and ~18–22 m thick) in the Permian Cedar Mesa Sandstone in Utah (Mountney, 2006), and aeolian–alluvial cycles (~2–15 m thick) in the Permian Ingleside Formation in Colorado (Pike and Sweet, 2018), which are interpreted to reflect the 100-kyr and 400-kyr eccentricity cycles. A previous study also suggested that the bioturbated zones and bounding surfaces in the Navajo Sandstone likely reflected orbital-scale pluvial episodes (Loope & Rowe, 2003). Although the Navajo Sandstone does not contain well-defined facies cycles or well-developed paleosols, in the Zion region the bounding surfaces appear to occur every ca. $2.9 \pm 0.9$ m of stratigraphic thickness (Fig. 2b-d; Supplementary Table). On the basis of existing chronological data (Dickinson and Gehrels, 2009; Dickinson et al., 2010), we estimated the duration of deposition of the Navajo Sandstone to be ca. 14.7–19.4 Myr and its thickness to be 300–700 m; thus, the average accumulation rate can be calculated as ca. 1.5–4.8 cm/kyr. Based on this estimated accumulation rate, the bounding surface of dune strata (every ~2.9 m) formed at intervals of 60–193 kyr, which is in agreement with the time-scale of the 100-kyr eccentricity cycle, consistent with previous studies. These lines of evidence, in conjunction with a comparison of reconstructed palaeo-wind directions and model-generated wind patterns, indicate that orbital-scale climate change may have influenced the development of longitudinal dune-fields in Pangaea." (Lines 258–274 of the revised manuscript).

Although there might be slight disagreement to referee's claim, we believe that our finding of periodicity in boundary surfaces can be linked to several cyclic climate forcings, one of which is orbital cycles. Nevertheless, we agree that aeolian sequences are influenced by both erosion and deposition, and so the preservation of orbital-scale climatic changes is not straightforward. Thus, we added the following text to the end of the section: "Although preservation of the palaeoenvironmental record is generally hampered by the erosion of aeolian deposits, it is likely that the Navajo Sandstone was deposited with a higher rate of sand supply than that of the present-day Sahara Desert (Kocurek, 2003), which may have enabled the preservation of orbital-scale palaeoclimatic records. Nevertheless, the formation mechanisms of bounding surfaces and their responses to orbital-scale climatic changes remain uncertain even in the Quaternary, due to the stochastic noise of deposition/erosion and sampling issues (e.g., Telfer et al., 2010; Leighton et al., 2014; Hesse, 2016; Thomas and Bailey, 2017). Thus, further investigation is required to test our hypothesis." (Lines 274–280 of the revised manuscript).

They detract from what is otherwise quite a clear story with a strong conclusion in which
the apparent conflict between climate models and field data is resolved. This is worthy of
publication. However, some parts should be rewritten (as indicated on PDF) to make clear
that the model predictions are indeed being tested.

According to the referee's comments, we added detail description of model
predictions by Rowe et al., (2007) and Winguth and Winguth (2013), and comparing
interpretation from our field results (Lines 229–257 of the revised manuscript).

Another weakness is the current arguments suggesting a role for vegetation in stabilizing
the dunes and causing sand accumulation seem ill-founded. There is no parallel in the
Sahara today between sand thickness and vegetation cover. Indeed, globally, where dunes
are vegetated or partly vegetated sand cover is thin.

As the referee pointed out, we recognized that vegetation plays a role in stabilizing
dune activity, but it is not certain about whether it works toward increasing sand
accumulation in the present-day Sahara Desert. So we deleted such arguments from main
text.

Appendix A should be elevated to the main paper. It is interesting and valuable support
for the interpretation of the dunes as being longitudinal in origin.

  According to the referee's comment, we moved some sentences of Appendix A to
the main text (Lines 163–175 in revised manuscript). We also revised Figure 2 and 3 to
show comparison with internal structure of modern longitudinal dune (revised figure,
**Figs. 2e, 3e**).

Conversely, I question the interpretation of some sites as being barchans dune deposits. I
think it is impossible today to find a site where barchans dunes form in a thick sediment
sequence with preservation potential. It is quite well documented that they occur where
sand supply is very low and quite often on hard surfaces which aid sand transport.
Furthermore, there is no modern parallel for contemporaneous and nearby barchans and
longitudinal dunes to have divergent orientations (figure 1). I think the explanation for your single slip-face orientations is most likely to be a sampling issue.

As the referee pointed out, we cannot rule out the possibility that the single slip-face orientations at the northernmost site is due to the sampling issue. The northernmost site, which might have been located on the margins of the palaeo-dune field, has a limited exposure with thin aeolian sequence due to the low sand supply at the time, making it difficult to interpret its dune morphology. Thus, we changed it to transverse dune according to the modern analogy. We revised interpretation sentence as follows: "In the northernmost area (~27°N), westerly winds dominated in summer during the eccentricity-modulated precession minimum, forming eastward- migrating transverse dunes." in Lines 256–257 of the revised manuscript. We also revised a sentence in Lines 225–227 as follows: "The northern area (palaeolatitude: ~24°–26°N) shows a bi-directional palaeo-wind pattern toward the SW and SE, whereas the northernmost area (palaeolatitude: ~27°N) shows a stronger influence of eastward palaeo-wind, although the possibility of sampling bias should be considered.".

**Response to other specific comments indicated in supplementary PDF**

Comment: I have noted minor issues of grammar, spelling and word usage on the manuscript.

We have revised grammar and spelling as suggested. The terms "overturning", "turnover", and "inversion" are changed to "alternation". The term "westward" is changed to "easterly". The term "precession" is changed to "eccentricity-modulated precession".

Comment at Line37-38: this may be true of Australia, S America and Africa/Arabia but is not true of the largest continents (relevant to this study?) of N America and Eurasia where the deserts and dunefields are at higher latitudes and dominated by westerly winds. The aridity is created in large part by the continental climate rather than the subtropical Hadley cell - or at least a combination of these.

We agree to referee's comments. So we revised the sentences in revised manuscript as follows, "Modern deserts are mostly developed in the subtropical high-pressure belt as a result of downwelling of the Hadley circulation, except for the interiors of Eurasia and

North America where a continental climate and monsoonal circulation are predominant."
in Line 37–39 of the revised manuscript.

Comment at Line92: exposure? outcrop? the dune field cannot be more extensive than
the strata from which it is inferred.

Marzolf (1988) estimated the minimum and maximum extent of the distribution of palaeo-dune fields of the Lower Jurassic sandstone. The minimum estimate is based on
the area enclosed by the zero isopachs of relatively continuous outcrops (Jordan, 1965);
the maximum estimate is based on the area excluding the Mogollon highlands, which may
interrupt the sand transport across the basin area during the Early Jurassic (Bilodeau and
Keith, 1986). The missing outcrop may not have survived subsequent erosions after lithologenesis. Based on this evidence, we revised the term as follows, "The estimated
size of the palaeo-dune field is ~625,000 km$^2$, which is 2.5 times larger than the size of
the remaining outcrop (Marzolf, 1988; Kocurek, 2003; Tape, 2005)." in Line 89–91 of
the revised manuscript, and added Marzolf (1988) and Kocurek (2003) to references.

Comment at Figure1: the inference of barchan dunes seems problematic. I don't think it
is possible today to find an example today of barchan dunes not aligned in the same
direction as nearby longitudinal dunes. Perhaps they were not coeval? However, barchan
dunes also form exclusively where sand supply is low, on non-sandy substrates and have very low likelihood of formation within depositional basins or long-term preservation.

As the referee points out, there is no example of barchan dunes which are not aligning
to the same direction as nearby longitudinal dunes. Barchan dunes are generally stretched
and connected to have formed longitudinal dunes, aligned in the same direction, for instance in the northern Taklamakan Desert. In addition, barchan dunes generally occur
where sand supply is low and are rarely preserved in the rock record (e.g., Lancaster,
2009). Thus, we changed interpretations to transverse dune. We revised interpretation
sentence as follows: "In the northernmost area (~27°N), westerly winds dominated in
summer during the eccentricity-modulated precession minimum, forming eastward- migrating transverse dunes." in Lines 256–257 of the revised manuscript. We also revised
a sentence in Lines 225–227 as follows: "The northern area (palaeolatitude: ~24°–26°N)
shows a bi-directional palaeo-wind pattern toward the SW and SE, whereas the northernmost area (palaeolatitude: ~27°N) shows a stronger influence of eastward palaeo-wind, although the possibility of sampling bias should be considered.". We also changed barchan dune symbol to transverse dune symbol in revised **Figure 1**.

Comment at Line234: since each cross-bed set will have been deposited on a timescale of days to years (?) I would think it would be more realistic to calculate the periodicity of cross-bed sets (= dunes). You should end up with roughly the same answer but it respects the depositional process. It doesn't, of course, address the question of the completeness of the stratigraphic record. See Telfer et al. (2010), Bailey and Thomas (2017) for modelling of deposition and erosion in longitudinal dunes. You have an accumulation of surviving beds, not all deposited beds.

As referee pointed out, longitudinal dunes might have been responded to annual and decadal wind variability; however, evidence of OSL dating of large-sized longitudinal dunes (e.g., Bristow et al., 2007) indicate much longer time-scale (up to centennial- to millennial-scale) of accumulation rate of surviving beds. Instead of calculating periodicity of cross-sets, we calculated the periodicity of bounding surface in the Zion region (every ca. 2.9 ±0.9 m of stratigraphic thickness; data-sets is shown in **Supplementary Table**).

We agree that aeolian sequences are influenced by both erosion and deposition, and so the preservation of orbital-scale climatic changes is not straightforward. However, given that previous studies have suggested a higher rate of sand supply in the Navajo Sandstone than in the present Sahara Desert (Kocurek, 2003), it is possible that the rate of sedimentation exceeded the rate of erosion in the dune fields at that time, and that a record of orbital-scale climatic changes may have been preserved. Thus, we added the following sentences in Lines 274–277 of the revised manuscript, such as, "Although preservation of the palaeoenvironmental record is generally hampered by the erosion of aeolian deposits, it is likely that the Navajo Sandstone was deposited with a higher rate of sand supply than that of the present-day Sahara Desert (Kocurek, 2003), which may have enabled the preservation of orbital-scale palaeoclimatic records. Nevertheless, the formation mechanisms of bounding surfaces and their responses to orbital-scale climatic changes remain uncertain even in the Quaternary, due to the stochastic noise of deposition/erosion and sampling issues (e.g., Telfer et al., 2010; Leighton et al., 2014; Hesse, 2016; Thomas and Bailey, 2017). Thus, further investigation is required to test our hypothesis.".

Comment at Line238: the Quaternary International Dune Atlas special edition (2016).   I don't think there is any Quaternary evidence to link episodes of dune activity to orbital frequencies. This is most likely because (1) dunes occur predominantly in arid areas already at or below the climatic threshold for activity (2) the dune record is compromised by partial erosion (the Sadler effect), see comment above).

As the referee pointed out, we recognized that Quaternary evidence does not support the dune activity to orbitally forced climatic change. Thomas and Bailey, 2017 suggested that there is no clear relationship between orbital insolation and dune-field activity in South Africa and Australian deserts. So we deleted almost all of sentences related to orbital-scale changes of dune-fields from Quaternary records from main text. We also deleted **Appendix B** and **Fig. B1**.

Comment at Line243: for reasons stated above, and because of the very poor age constraints, I think this section is much too speculative. I think it is largely unnecessary as well.

As stated above, we extensively revised the discussion of orbital cycles. But we retained some discussion of the possibility of orbital-scale climatic change being recorded in aeolian depositional sequences in subtropical Pangaea, using evidence from previous studies.

Comment at Line245: this is much too speculative, and even contradicts the paragraph above where you calculate periodicity from 42 kyrs upwards.

Winguth and Winguth (2013) documented eccentricity-modulated precession cycle affect to the climatic changes in tropical and subtropical area of Pangaea supercontinent. Calculated periodicities of boundary surface in the Zion region is every ca. 2.9 ±0.9 m of stratigraphic thickness. Based on the existing chronological data, estimated average accumulation rate of 1.5–4.8 cm/kyr yield bounding surface occurred every 60–193 kyr. These values are in agreement with the time-scale of the 100-kyr eccentricity cycle. The periodicity of the eccentricity cycle we have estimated from the strata in Zion is overall

consistent with the existing evidence from the previous studies (Loopw and Rowe, 2003; Mountney, 2006; Hassan et al., 2018; Pike and Sweet, 2018) (Lines 264–270 of the revised manuscript).

Nevertheless, we agree to the reviewer's claim such that aeolian sequences are influenced by both erosion and deposition, and so the preservation of orbital-scale
climatic changes is not straightforward. Thus, we added the following text to the end of the section: "Nevertheless, the formation mechanisms of bounding surfaces and their responses to orbital-scale climatic changes remain uncertain even in the Quaternary, due to the stochastic noise of deposition/erosion and sampling issues (e.g., Telfer et al., 2010; Leighton et al., 2014; Hesse, 2016; Thomas and Bailey, 2017). Thus, further investigation
is required to test our hypothesis." (Lines 277–280 of the revised manuscript).

Comment at Line250: are there palaesols preserved in the sequence? the link to the next sentence needs to be stronger and the information more explicit. What type of nodules?
what type of trace fossils?"
Comment at Line253 "nodular layers": of what?"

Although the Navajo Sandstone does not contain well-developed palaeosols in the Zion region, the bounding surfaces appear to occur every ca. 2.9 ±0.9 m of stratigraphic
thickness. In addition, evidence of trace fossils and spherical nodules are abundantly observed within eolian dune sandstone of Navajo Sandstone in south-central Utah (Chan and Archer, 2000; Loope and Rowe, 2003). This is probably formed by changes in groundwater level and evaporative concentration of calcite. Trace fossils is probably invertebrate burrows (made by beetles and arachnids), which may be actively formed
during long-lived pluvial intervals (Ekdale et al., 2007).

We revised corresponding sentences as follows: "The movement of dune sand was probably stabilized by intense summer rainfall and resulting higher groundwater table and enhanced vegetation (Kocurek, 2003; Durán and Herrmann, 2006; Hesse and Simpson, 2006), which seems consistent with the development of bounding surfaces and evidence
of trace fossils (invertebrate burrows) within dune slip-faces in south-central Utah (Chan and Archer, 2000; Loope and Rowe, 2003; Ekdale et al., 2007)." (Lines 242–246 of the revised manuscript). We also added Durán and Herrmann, 2006 and Loope and Rowe (2003) to the references.

Comment at Line252: If you look at Saharan ergs today, they often have distinct patterns of sand thickness that are unrelated to vegetation cover (because there isn't any), but related to sand drift direction and accumulation (i.e. thicker towards the downwind edge). This doesn't seem to occur in more vegetated dune fields (southern Africa, Australia) where sand transport is much more restricted.

As stated above, we deleted arguments for the role of vegetation in increasing sand accumulation from main text.

Comment at Line259: this paragraph is somewhat confusing because there is not enough distinction between model results (predictions) and field results (empirical evidence to test the predictions)
Comment at Line265: same - this is somewhat determinative. Given that you set up the study to resolve the incompatibility of previous model results and field results you should be careful to present your new results as a test of the model predictions.

According to the referee's comments, we added detail description of model predictions by Winguth and Winguth (2013), and comparing interpretation from our field results (Lines 234–257 of the revised manuscript). We also revised Fig 4.

Comment at Line278-282: I think you need to clearly separate what those authors say and any new interpretations you place on their data. For example, Thomas and Burrough do not interpret their record in terms of insolation and this sounds very unlike Thomas' view of these Kalahari records.

As stated above, we have deleted almost all of sentences related to orbital-scale changes of dune-fields from Quaternary records.

Comment at Line300: not convinced

As stated above, we retained some discussion of the possibility of orbital-scale climatic change being recorded in aeolian depositional sequences in subtropical Pangaea, using evidence from previous studies. We also added following sentences in the end of conclusion, such as: "The results also indicate the influence of orbitally induced climate change on longitudinal dune development in subtropical Pangaea, although further chronological and sedimentological studies are required to test this hypothesis." In Lines 304–306 of the revised manuscript.

Comment at Lines 310-312: I think not. Note that the evidence for dune accumulation in the Early Holocene southern Sahara does not carry evidence of great sand thickness.

As stated above, we recognized that role of vegetation plays a role in stabilizing dune activity, but it is not certain about whether it works toward increasing sand accumulation in the Sahara. So we deleted such arguments from main text.

Comment at Appendix A: I think this is worthy of inclusion in the results section of the main paper as convincing evidence for the dune architecture.

We moved some sentences of **Appendix A** to the main text (Lines 185–187 in revised manuscript). We also revised Figure 2 and 3 to show comparison with internal structure of modern longitudinal dune (revised figures, **Figs. 2e, 3e**).

Comment at Appendix B of Line342: what do more recent studies say?
Comment at Line343: when?
Comment at Line344: do you mean MIS2? MIS2-4? LGM only?
Comment at Line345: the timing is important because, as Thomas and Burrough 2016 show, these areas did not show enhanced LGM activity but enhanced activity before and after the LGM. The SW Kalahari, on the other hand experienced greatest activity (number of dune ages) around 10-12 ka.
Comment at Line347: in the sense that you are talking about patterns over 10^3 or 10^4 years, but not with any strong evidence of orbital forcing, according to Thomas in various publications.

As stated above, we have deleted almost all of sentences related to orbital-scale changes of dune-fields from Quaternary records. **Appendix B** and **Fig. B1** are also deleted.

---

## Author Response (AR2)

**Responses to comments from Anonymous Referee #2 and Editor**

We are grateful to the referee #2 for suggestions. Below, we give our responses in turn following each comment, with the reviewers' comments being in with underline and our responses being in without underline.

line 17 - delete 'of'
We have corrected it as suggested.

line 18 - change 'to be' to 'were'
We have corrected it as suggested.

line 98 - change 'From the' to 'The'
We have corrected it as suggested.

line 100 - change 'as' to 'of'
We have corrected it as suggested.

The sentence beginning line 269 still pushes the idea of orbital control of deposition too far. The following paragraph is correctly circumspect but this sentence encourages the reader to believe that the authors truly believe that eccentricity controls the timing of aeolian deposition. The chronology, being so imprecise, and the use of averages may give ballpark agreement but I think this is likely to be no more than coincidence.

According to the referee's suggestion, we revised the sentence in lines 269-272 in revised manuscript, such as "Based on this estimated accumulation rate, the bounding surface of dune strata (every ~2.9 m) likely formed at intervals of 60–193 kyr, which is in agreement with the time-scale of the 100-kyr eccentricity cycle, consistent with previous studies, although further chronological and sedimentological studies are required to test this hypothesis." (Red characters indicate revisions).